# Dynamic Handover:
# Throw and Catch with Bimanual Hands

**Binghao Huang**[*,1]    **Yuanpei Chen**[*,2]    **Tianyu Wang**[1]    **Yuzhe Qin**[1]
**Yaodong Yang**[2]    **Nikolay Atanasov**[1]    **Xiaolong Wang**[1]

UC San Diego[1]    Peking University[2]

**Abstract:** Humans throw and catch objects all the time. However, such a seemingly common skill introduces a lot of challenges for robots to achieve: The robots need to operate such dynamic actions at high-speed, collaborate precisely, and interact with diverse objects. In this paper, we design a system with two multi-finger hands attached to robot arms to solve this problem. We train our system using Multi-Agent Reinforcement Learning in simulation and perform Sim2Real transfer to deploy on the real robots. To overcome the Sim2Real gap, we provide multiple novel algorithm designs including learning a trajectory prediction model for the object. Such a model can help the robot catcher has a real-time estimation of where the object will be heading, and then react accordingly. We conduct our experiments with multiple objects in the real-world system, and show significant improvements over multiple baselines. Our project page is available at `https://binghao-huang.github.io/dynamic_handover/`.

**Keywords:** Bimanual Dexterous Manipulation, Sim-to-Real Transfer

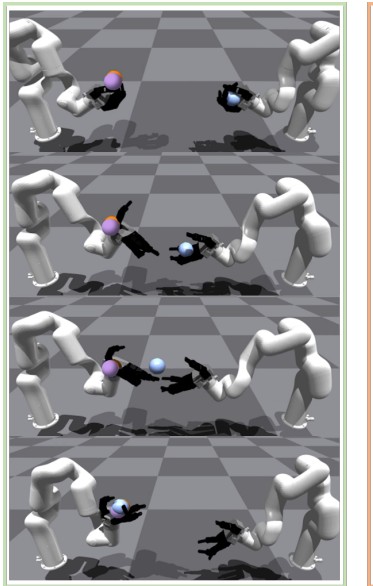
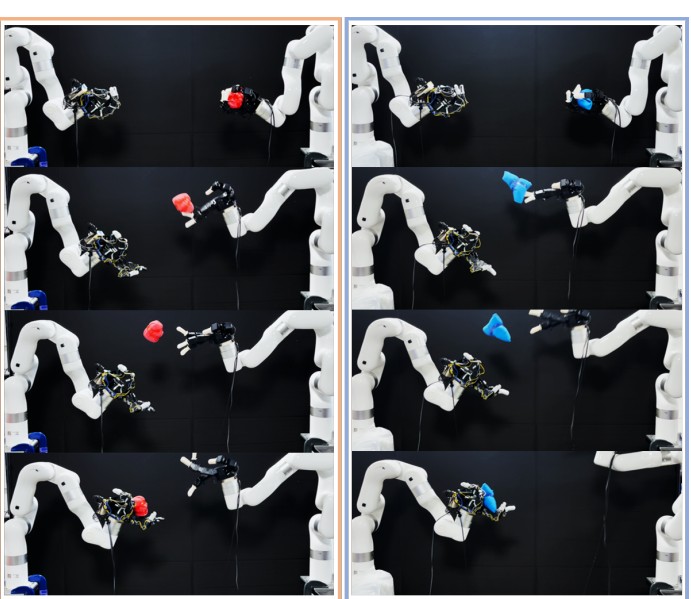

Training in Simulation          Test in Real Robot System

Figure 1: We propose **Dynamic Handover**, a new bimanual dexterous hands system designed for throwing and catching tasks. The system consists of two Allegro Hands, each individually attached to a separate XArm robot, arranged in a facing configuration. Using multi-agent reinforcement learning, we train policies in a simulation environment and subsequently transfer them to the real world.

---

* Indicates equal contribution.

7th Conference on Robot Learning (CoRL 2023), Atlanta, USA.

# 1 Introduction

Human's ability to throw and catch objects exists commonly in both sports (e.g., baseball) as well as casually passing through items in daily life. In the context of handing over objects, enabling the robots to obtain such bimanual manipulation skills not only improves efficiency, but also extends the workspace of the robots by a large margin outside the robots' kinematic range. Importantly, it provides a contact-free and safer solution for robot interactions, since direct handing objects over can cause collisions between two manipulators, which brings damage. To enable robust manipulation of diverse objects, we propose to use two multi-finger robot hands as the thrower and the catcher.

Learning such skills brings multiple challenges: (i) *Precise, high-speed dynamic action*: Both robot hands will need to obtain dynamic extrinsic dexterity at high-speed and operate precisely at the same time. A small error or a mismatch of coordination will easily lead to failure. (ii) *Diverse objects*: To train a policy that is applicable to diverse objects makes the problem even more challenging. Traditional approach with planning based on full object state estimation can introduce new errors and is hard to generalize to unseen objects with limited training data; (iii) *High-dimensional action space*: Learning bimanual manipulation with two multi-finger hands introduce a high-dimensional state and action space, which increases the difficulty for optimization.

In this paper, we propose a new system and a new method for throwing an object with one hand and catching it with the other hand in a coordinated manner. Our hardware system includes two xArm robots attached with two Allegro Robot Hands (each with 16 degrees of freedom) in the end separately as shown in Fig. 2. The two hands will need to operate collaboratively at the same time. To solve this problem, we propose an approach leveraging Reinforcement Learning to train a policy in simulation and then perform Sim2Real transfer on the real robots. Specifically, to solve the above challenges, we introduce three key technical contributions:

- **Multi-Agent Reinforcement Learning (MARL).** Dexterous manipulation with RL has been extensively studied for learning robust and generalizable skills [1, 2, 3]. However, it has rarely been explored for dynamic manipulation with bimanual hands. We propose to tackle this problem as a multi-agent problem, with each hand being one agent. This helps improve the coordination of two hands during manipulation which allows better Sim2Real transfer. Meanwhile, we train the policy with multiple diverse objects at the same time, which allows generalization to unseen objects.
- **Dynamic dexterity.** Instead of explicitly estimating the dynamic properties and modeling them, using RL with randomization allows the network to learn to handle diverse dynamics. We apply various physical property randomization during training, including randomization in friction, inertia, the object's center of mass, and contact force. To enhance the stability of the thrower, we also investigate how the quality of the initial grasp of the object can affect the performance.
- **Object trajectory prediction.** While we can perform various randomization approaches for training, there will always be a physics gap when transferring to the real robot. To further increase the robustness for catching, we introduce a model to predict the object's future trajectory and destination ahead in real time. Instead of pre-defining the object destination and asking both hands to follow, the catch policy will take the predicted object's destination position as input in a close-loop manner. This allows flexible real-time adjustment of the catch policy.

We conduct both simulation and real-world experiments. The results demonstrate that our method enables the hands to successfully throw and catch the object, and surpass baselines by a large margin. We observe applying MARL in training effectively reduces the sim2real gap, as using partial observations for each agent improves the robustness of each policy. Our prediction module offers extra flexibility to help the robot hand to adjust in real-time and accurately catch the object. To our knowledge, this is the first work that enables bimanual dynamic multi-finger hands manipulation.

# 2 Related Work

**Multi-Agent Reinforcement Learning.** Multi-Agent Reinforcement Learning (MARL) is a powerful approach for complex tasks involving multiple interacting agents. In MARL, agents learn decision-making based on interactions with others and the environment [4, 5, 6, 7, 8]. Multi-Agent Proximal

Policy Optimization (MAPPO) [9] extends policy gradient algorithms into multi-agent scenarios by addressing challenges such as non-stationarity, coordination, and credit assignment. In robotics, MARL has been applied to cooperative navigation [10, 11], UAV swarm [12, 13], and distributed manipulation [14, 15]. Similar to us, Zhang *et al*. [16] employ disentangled attention within MARL for bimanual tasks, while Li *et al*. [17] use symmetry-aware actor-critic for efficient handover. Both studies were conducted exclusively in simulation without complex dynamics, allowing for end2end MARL training. In contrast, we propose a multi-stage training pipeline for MARL to overcome sim2real transfer challenges.

**Dynamic Manipulation.** Different from common manipulation tasks like grasping and opening drawers, dynamic manipulation abandons the quasi-static assumption of interaction. It leverages object dynamics, such as inertia and momentum, to handle tasks requiring high-speed actions and extended robot workspace, like throwing and catching [18, 19, 20, 21]. Traditional systems for such tasks often rely on handcrafted models of system dynamics, which may fall short when dealing with difficult-to-estimate parameters or new objects. To address this issue, recent works [22, 23, 24, 25, 26] employ data-driven approaches to optimize control commands using partial dynamics models. For example, Chi *et al*. [25] propose an iterative residual policy to solve tasks with complex dynamics; Zeng *et al*. [24] use end-to-end training to learn stable grasps that generate predictable throws. However, these works focus on low-DoF manipulators rather than high-DoF dexterous hands, which introduce additional complexities due to intricate hand-object interactions. [27] uses Population Based Training (PBT) to train the throws and catches using dual hands. However, while our methodologies primarily focus on sim2real transfer, [27] does not include the real-world experiments. In this study, we use a learning-based approach to tackle dynamic problems, specifically throwing and catching, with multi-finger dexterous hands. We explore the impact of initial dexterous grasps on throwing performance and investigate strategies to bridge the sim2real gap in the context of dynamic manipulation.

**Bimanual Dexterous Manipulation.** In recent years, the robotics community has increasingly focused on dexterous manipulation due to its great flexibility and human-like dexterity. Researchers have developed methods using dexterous hands for tasks such as grasping [28, 29, 30, 31, 32], in-hand rotating [33, 34, 35, 36, 37], and manipulating deformable objects [38, 39, 40, 41]. Similar to us, DexPoint [42] achieves generalizable manipulation for grasping and door opening by training on multiple objects with a Allegro hand. However, complex tasks like throwing and catching objects require a bimanual robot system to achieve human-level manipulation skills. Researchers have investigated bimanual manipulation through task planning [43, 44, 45], and reinforcement learning [46, 47, 48]. However, most previous work focused on using two parallel jaw grippers for quasi-static interaction, leaving dexterous bimanual manipulation largely unexplored. Few studies have delved into this area, mainly in simulation without real robot validation [49, 50, 51]. In this paper, we take a step forward by developing a bimanual dexterous manipulation system capable of throwing and catching various objects. Our work also demonstrates that simulation training without real-world data can still tackle this challenging task even with great sim2real gap.

## 3 System Setup

**Task Description.** In this work, we focus on the bimanual Catching and Throwing task with two dexterous robot hand. This task involves two robot agents: (i) a thrower robot agent (Figure 2 right) that needs to execute swift movements to toss the grasped object towards the other side, and (ii) a catcher agent (Figure 2 left) that needs to react dynamically to catch the airborne object. This task is important because it enables the catcher robot to access objects beyond its kinematic range by leveraging the object's momentum imparted by the thrower. It also serves as a good test-bed for evaluating the coordination and performance of bimanual systems in high-speed, real-time scenarios.

**Real-world Setup.** We construct a bimanual system for executing our throwing and catching task, as depicted in Figure 2. The system includes two arm-hand subsystems and a RealSense D435 camera. Each arm-hand subsystem features a 6-DoF XArm-6 robot arm paired with a 16-DoF Allegro Hand, culminating in a 44-DoF system. To create a closed-loop policy, we use a RealSense camera to

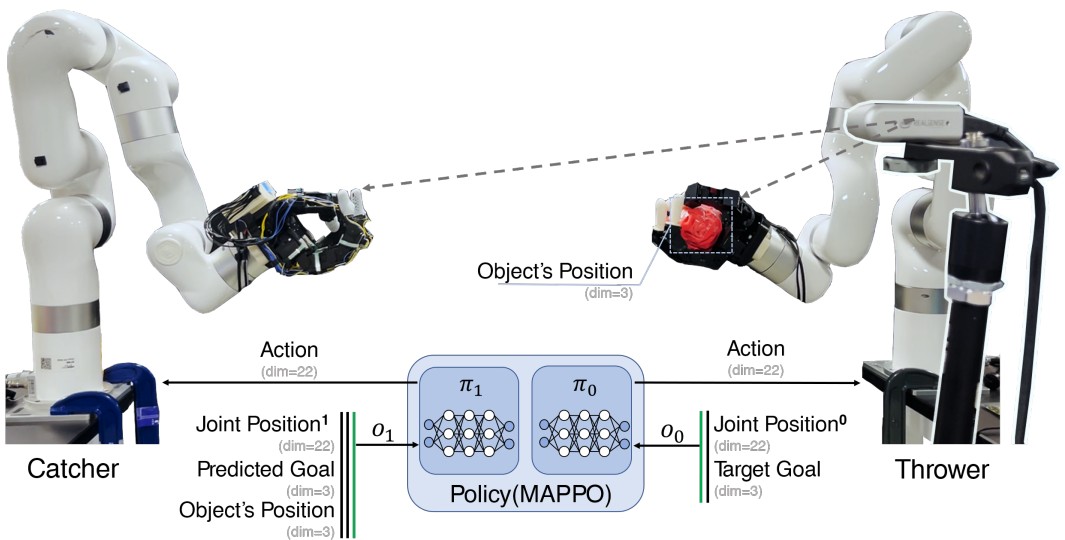

Figure 2: **Real Robot System:** We employ two Allegro Hands, each individually mounted on separate XArm-6 robots, arranged in a face-to-face configuration. We incorporate a RealSense D435 camera for real-time object position tracking, which is oriented towards the working space. We use k prior states in observation.

capture the real-time position of objects within the catcher robot's frame of reference. For observation, distinct feedback mechanisms are provided for each agent, as depicted at the bottom of Figure 2. The thrower (Figure 2: right) depends exclusively on its own proprioceptive data, while the catcher (Figure 2: left) obtains feedback from not only its own proprioception but also the object's real-time positions estimated by the camera. In other words, the thrower operates based on its current state, whereas the catcher necessitates both proprioceptive and visual input to dynamically and interactively perform catching actions. Further information regarding this system's implementation can be found in the Appendix.

**Simulation Setup.** In this work, we use the IsaacGym physical simulator [52] for training our throwing and catching task. The simulation setup is shown on the left side of Figure 1. The simulation frequency is set at 120Hz while the control frequency is 20Hz. We train the end-to-end reinforcement learning policy in the simulated environment and then transfer the policy to the real world. Further information regarding the details about the simulation setup (e.g. policy architecture) can be found in the Appendix.

**Action Space.** The policy outputs a 22-dimensional PD control target, with the first six dimensions corresponding to XArm-6 and the remaining 16 dimensions corresponding to Allegro hand. For the XArm-6, we employ delta joint positions as the control target, while for the Allegro hand, we utilize absolute joint positions as the control target. This design choice is made to avoid jerky motion of robot arm for safety reasons, while still allowing the hand to swiftly react and release its grasp to throw the object. In our experiments, we find that controlling only the second and third joints of the robot arm and keeping the other four joints fixed results in a more effective and safer policy. Therefore, the action space consists of an 18-dimensional target for the thrower and a 22-dimensional target for the catcher.

## 4 Learning Bimanual Dexterous Hands Policy

Catching an object in mid-air poses significant difficulties due to the high-speed requirement. First, object's real-time velocity and anticipated trajectory must be taken into account in order for the catcher to determine its movement. Second, even though the thrower can consistently toss the object toward the pre-defined target goal in simulated environment, the policy transfer to the real world is imperfect due to the substantial dynamics gap between simulation and real physical. Consequently, there is a discrepancy between the pre-defined throwing goal and the object's actual destination. In light of these two challenges, a goal estimator becomes crucial for predicting the thrower's actual

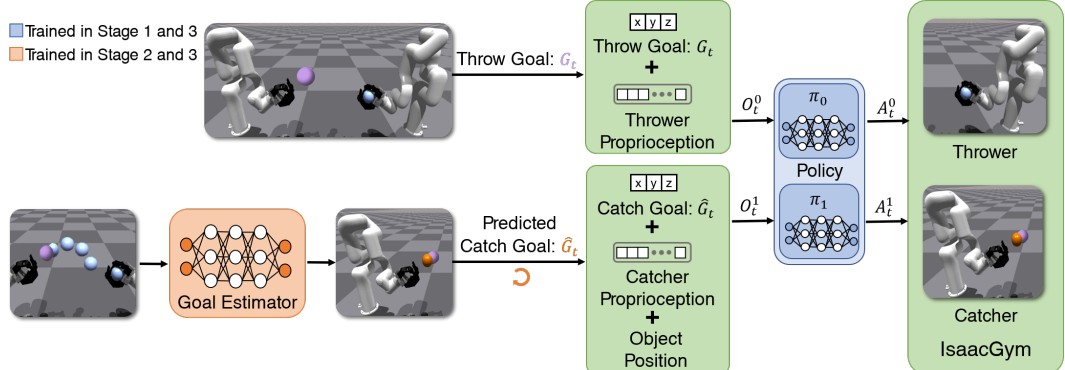

Figure 3: **Joint End2End Learning:** The two agents receive input from both their own observations and the catcher agent additionally receives the predicted catching position. The goal estimator takes past 20 frames of the object's positions as input and predicts the catch goal for each time step. We use a violet ball to represent the pre-defined goal for the throwing. The orange ball represents the predicted goal for the catcher to catch during the throwing task. The blue ball represents the object that is currently been manipulated.

destination instead of the predetermined target goal. This allows the catcher to move based on the forecasted object destination and successfully catch it.

To achieve this, we introduce a novel three-stage training pipeline for learning bimanual throwing and catching. (i) In the first stage, we train a base policy using Multi-Agent RL to tackle the task, with the catcher observing the pre-defined throwing goal. The policy trained during this stage is expected to perform well within the simulator but may hardly transfer to the real world. (ii) Next, we freeze the base policy and train a goal estimator through supervised learning, using the rollout trajectory of the base policy as training data. (iii) Finally, we replace the pre-defined throwing goal in the catcher's observation with the estimated goal and unfreeze the policy for fine-tuning both the base policy and the goal estimator in an end-to-end fashion. The refined policy is expected to bridge the dynamics gap between simulation and reality with the predicted object's future trajectory.

## 4.1 Stage1: Multi-Agent Reinforcement Learning.

We employ the Multi-Agent Proximal Policy Optimization (MAPPO) [9] in a non-parameter sharing way to train the thrower and the catcher to obtain basic policies in the first stage. MAPPO is an application of the PPO algorithm to multi-agent settings. It leverages centralized training with decentralized execution, allowing each robot agents to efficiently accomplish the cooperative task using partial observations.

As shown at the bottom of Figure 2, the observations for the thrower and catcher in MAPPO training is not identical. The thrower's policy, denoted as $\pi_0$, receives its proprioception and a pre-defined target position for throwing. In contrast, the catcher's policy, denoted as $\pi_1$, takes as input its proprioception, the pre-defined goal position and the current position of the object for catching. To satisfy MAPPO's requirement for equal input dimensions across agents, we pad zeros to the thrower's input for dimension alignment. Besides, we include observations from past $k$ frames as input for both policies to provide temporal information. In our implementation, we set $k = 2$.

Given the object position $\boldsymbol{p}_t$, target goal position $\mathcal{G}_t$, the velocity of the object $\boldsymbol{v}$, the unit direction vector from thrower to catcher $\hat{\boldsymbol{u}}$, and robot joint torque $\boldsymbol{\tau}$, we design the reward function using three components: (i) distance between object and throwing goal; (ii) object velocity projected in the direction from thrower to catcher; (iii) robot joint torque. The final reward $r$ can be computed as $r = r_{dis} + r_{linvel} + r_{torque}$, where $r_{dis} = exp(-20 * (\boldsymbol{p}_t - \mathcal{G}_t))$ represents the distance, $r_{linvel} = clamp(\boldsymbol{v} \cdot \hat{\boldsymbol{u}}, -0.1, 0.1)$ denotes the object's velocity towards the catcher, and $r_{torque} = -0.003 * \|\boldsymbol{\tau}\|_2^2$ corresponds to the torque penalty.

## 4.2 Stage2: Goal Estimator Learning.

After training the basic policies, the next step involves freezing the basic policies and training a goal estimator (Orange Block in Figure 3) to predict the goal for the catcher based on the trajectory of

the object. This is a crucial step due to the sim2real gap, which implies that although the thrower may consistently hit the goal in simulation, it is unlikely to achieve the same level of accuracy in the real world. Therefore, predicting the actual goal based on the object's trajectory becomes essential for improving sim-to-real transfer. In this stage, we utilize the historical positions of the object over a span of $k$ frames as input to the goal estimator. The output of the goal estimator is the predicted 3D position of the goal, which provides crucial information for the catcher to anticipate the intended catching point and enhance the coordination between the two hands. We use Adam to optimize the L2 distance between the position of the predicted goal and the thrower's goal until convergence:

$$\mathcal{L}(\omega) = \|\omega(\mathbf{p}^1_{t-k:t}) - \mathcal{G}^0_t\|^2 \tag{1}$$

where $\mathbf{p}^1_{t-k:t}$ is the position of the object from $t - k$ to $t$ frames, $\mathcal{G}^0_t$ is the thrower's goal, and the 0 and 1 represent the thrower and catcher respectively. It is important to highlight that in the previous training stage, the object's landing point was primarily influenced by the thrower's goal. This is because the thrower, operating in the simulation environment, had the ability to consistently hit the specified goal in the simulation.

### 4.3 Stage3: End2End Joint Learning.

In stage 1, the catcher's base policy uses a pre-defined throw goal in its observation. In this stage, we replace the pre-defined goal with the predicted one from the goal estimator. However, the distribution shift brought by this replacement can result in compounding errors. To address this issue, we jointly fine-tune the goal estimator and the policy network in this stage, as visualized in Figure 3, allowing the catcher to adapt to the goal estimator. For example, when the goal estimator is inaccurate, the policy will not depend solely on it for decision-making. This joint training approach helps reduce compounding errors when integrating the goal estimator with the policy.

## 5 Experiments

**Evaluation Criterion.** To evaluate the performance of the trained policies for throwing and catching, we consider several metrics as follow:

*(i) Success Rate(SR):* It is calculated as the ratio of successful throws and catches to the total attempts. A better policy will lead to a higher Success Rate.

*(ii) Hit Rate(HR):* This metric is defined as the proportion of objects that successfully hit the hand palm of catcher. A better policy and goal estimator will lead to a higher Hit Rate.

**Training and Dataset.** During the training process, we utlize three different objects: a ball, a cube, and a rod, which are three typical geometries for robot manipulation. At the beginning of each episode, we randomly select one object for training. For simulation experiments, we expand the object set to include additional objects to evaluate the generalizability of our policy to novel objects. The objects used in the simulation experiments are depicted in Figure 5(a) and Figure 5(b). We intentionally incorporate a rod during the training procedure, as we find this leads to a more robust policy in the training. Specifically, we observed that the simultaneous movement of all fingers when throwing an object led to more robust results, whereas when using balls/square objects, policy often used only a few fingers. Therefore, we designed the shape of the rod in training. If the policy wants to throw the rope stably, it must learn to use all of its fingers in throwing, so it can make our policy more robust. Further discussion about using the rope for training can be found in the Appendix. In the real-world experiments, we employ sandbags in three different shapes for throwing, as shown in Figure 5(c): a ball, a cylinder, and a triangle prism.

**Baselines.** In this paper, we compare our method with the following baselines:

*(i) Open-Loop Policy:* We employ a pre-defined trajectory for the bimanual hand-arm system to execute the throwing and catching task. The trajectories are collected on the real robot using kinesthetic teaching and replayed later without considering feedback or adjustment during execution.

| Settings | Known Obj. | Novel Obj. |
|---|---|---|
| w/o Multi-Agent | 0.89 ± 0.07 | 0.24 ± 0.05 |
| w/o Goal Estimation | 0.88 ± 0.04 | 0.22 ± 0.04 |
| w/o Both | 0.93 ± 0.07 | 0.12 ± 0.06 |
| Ours | **0.95 ± 0.07** | **0.37 ± 0.04** |

Table 1: **Ablation Study in Simulation:** Success Rate of throwing and catching task on different objects in simulation. We use **11** trained objects and **14** novel objects. The results are averaged on 5 seeds, each seed has 100 trails.

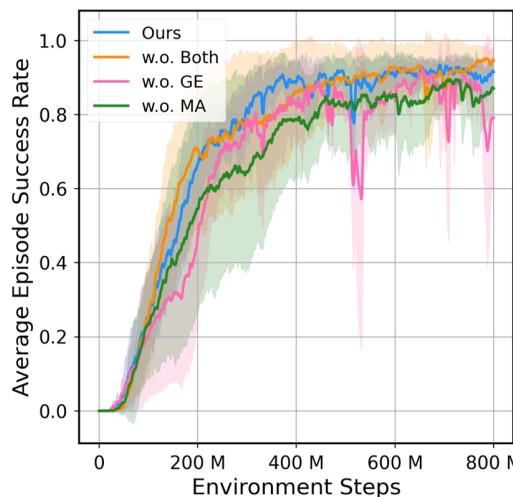

Figure 4: **Training Curves.** The plot shows multi-object training curves of our method and 3 baselines.

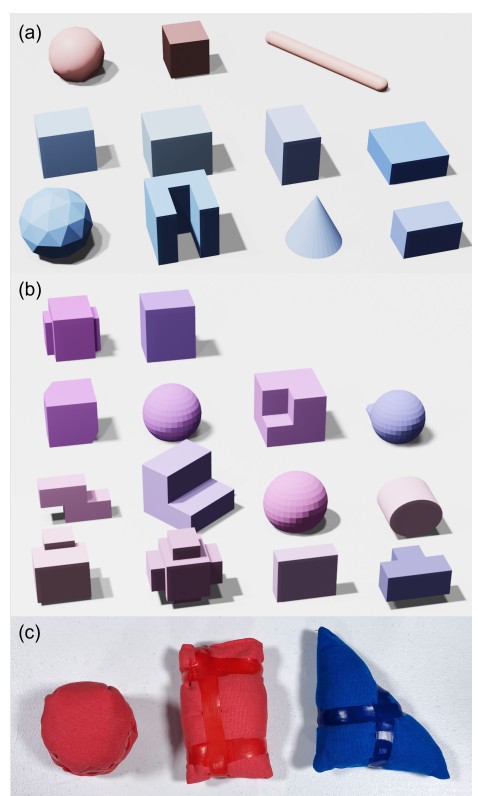

Figure 5: **Objects Sets.** (a) Training objects. (b) Additional objects in evaluation. (c) Real-world objects.

*(ii) Without Multi-Agent:* We train our policy using PPO instead of using MAPPO. However, we still incorporate goal estimation during the learning process. Under this setup, both agents share the same observation.

*(iii) Without Goal Estimation:* We restrict our learning process to the first stage in Section 4 with MAPPO algorithm and evaluate the policies without goal estimation.

*(iv) Without Both:* We restrict our learning process to the first stage in Section 4 with PPO algorithm and evaluate the policies without goal estimation.

## 5.1 Results in Simulation

We conduct an analysis of our method with three baselines in the simulation environment. Figure 4 shows the training curve of four methods. Table 1 presents the success rates for two categories: Known Objects and Novel Objects. Our findings can be summarized as follows.

First, for the test on known objects experiment, we observe that the baseline (Without Both) outperforms the other three methods, including ours. One possible reason for this is that the policy without multi-agent coordination has access to full observations of both hands and the arm system, as well as the ground-truth object position. As a result, the policy can easily overfit to a specific point and successfully solve the task. However, it is important to note that this baseline policy may have lower generalization capabilities when it comes to novel objects or uncertain parameters. In the real world, we encounter noise and uncertainties, and obtaining the ground-truth object position is not feasible. Therefore, the policy with full observations demonstrates lower transferability from simulation to the real world, as we validate in Section 5.3.

Secondly, in the novel objects experiment, although all methods show a significant drop in performance, our method outperforms the baselines. The utilization of multi-agent reinforcement learning and goal estimation proves beneficial for accomplishing the throwing and catching task. Both the

| Settings | Ball | | Cylinder | | Triangle | |
|---|---|---|---|---|---|---|
| | HR | SR | HR | SR | HR | SR |
| Open-Loop | $0.60 \pm 0.12$ | $0.13 \pm 0.12$ | $0.47 \pm 0.12$ | $0.13 \pm 0.12$ | $0.27 \pm 0.12$ | $0.07 \pm 0.12$ |
| w/o Multi-Agent | $0.73 \pm 0.31$ | $0.40 \pm 0.20$ | $0.53 \pm 0.31$ | $0.20 \pm 0.00$ | $0.47 \pm 0.12$ | $0.20 \pm 0.20$ |
| w/o Goal Estimation | $0.60 \pm 0.20$ | $0.33 \pm 0.23$ | $0.67 \pm 0.31$ | $0.40 \pm 0.34$ | $0.40 \pm 0.20$ | $0.13 \pm 0.23$ |
| w/o Both | $0.47 \pm 0.13$ | $0.12 \pm 0.12$ | $0.40 \pm 0.20$ | $0.07 \pm 0.12$ | $0.20 \pm 0.20$ | $0.00 \pm 0.00$ |
| Ours | $\mathbf{0.93 \pm 0.12}$ | $\mathbf{0.60 \pm 0.20}$ | $\mathbf{0.80 \pm 0.20}$ | $\mathbf{0.53 \pm 0.12}$ | $\mathbf{0.86 \pm 0.12}$ | $\mathbf{0.33 \pm 0.12}$ |

Table 2: **Ablation Study in Real-World:** Performance of throwing and catching task on 3 different unknown objects in real robot platform. Objects are made of sandbags with the same mass but different shapes. The results are averaged on 3 seeds with 5 trails for each. The two terms stand for:(i) Hit Rate(HR): This metric is defined as the proportion of objects that successfully hit the hand palm of catcher. A better policy and goal estimator will lead to a higher Hit Rate. (ii) Success Rate(SR): It is calculated as the ratio of successful throws and catches to the total attempts. A better policy will lead to a higher Success Rate.

thrower and catcher agents receive observations from their respective perspectives, enabling them to perform their tasks cohesively. Furthermore, goal estimation assists the catcher in predicting the landing point of the objects based on historical positions. This feature helps mitigate the impact of unpredictable parameters during the manipulation process, such as friction, unexpected collisions, and other dynamic factors.

## 5.2 Why MAPPO Outperform PPO in the Dynamic Handover?

Commonly, the single-agent setting should be strictly easier than the multi-agent setting, since both thrower and catcher have full access to the states of both sides. However, our result suggests otherwise which seems rather counter-intuitive. So why MAPPO outperform PPO in our system? We agree that single-agent RL might be easier than the multi-agent RL in training, so the performance of the single-agent RL and the multi-agent RL in the trained object are similar. But as single-agent RL can access more information, they are more likely to overfit into the environment, so they will be harder to generalize than multi-agent, which is why multi-agent performs well in novel objects in simulation and the sim2real transfer.

## 5.3 Results in Real World

We perform sim-to-real experiments to assess the performance of our method and two baselines on a real-robot platform. As depicted in Figure 2, we deploy multi-agent reinforcement learning policies on the real robot agents, with both agents controlled by the same host. The task execution sequence is visualized on the second and third columns in Figure 1. Further implementation details and communication methods can be found in the Appendix.

The results of our real-world evaluation are presented in Table 2. We successfully transfer our multi-agent reinforcement policy to the real robot system with a reasonable success rate after performing system identification to align the PD controllers between the simulation and the real robot. Our method outperforms the baseline methods, indicating the effectiveness of multi-agent reinforcement learning (MARL) and goal estimation in real robot experiments. These components provide benefits in dealing with various unpredictable factors encountered in the real-world setting, leading to improved performance and robustness. We also notice that the success rates achieved in real-world experiments are lower than the hit rate. This is primarily attributed to occasional challenges encountered during the grasping phase of the catcher. In some cases, the catcher may fail to firmly grasp the object before it bounces back, leading to the object slipping off the robot's hand.

## 6 Conclusion and Limitation

**Limitation** In our study, we acknowledge that the use of objects with low restitution may not fully capture the challenges faced in real-life scenarios where objects often have higher restitution. Objects with higher restitution tend to bounce back upon impact, making it more difficult to catch them smoothly without collisions.

**Conclusion** We present Dynamic Handover, a system capable of throwing and catching with bimanual hands. Through the use of multi-agent reinforcement learning and goal estimation, our system demonstrates the ability to achieve successful throw and catch in both simulation and real-world environments. We find that the goal estimation aids in mitigating the effects of unpredictable parameters and enhances the overall stability to bridge the large dynamics gap between sim and real.

**Acknowledgment.** This work was supported, in part, by the Qualcomm Innovation Fellowship, and the Technology Innovation Program (20018112, Development of autonomous manipulation and gripping technology using imitation learning based on visual and tactile sensing) funded by the Ministry of Trade, Industry & Energy (MOTIE), Korea.

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

# A  Detailed Implementation of Real Robot System

**Bimanual Hands System.** For our system, we have developed a ROS-based pipeline that operates at a control frequency of 20Hz. This pipeline serves as the foundation for controlling our setup, enabling efficient communication and coordination between the different components. In our configuration, the Arm-Hand subsystems are controlled by a single policy utilizing multiple agents. This unified policy governs the actions of both subsystems, promoting synchronized and collaborative behavior in our setup. To achieve this, we control the motion of the robotic arms through Modbus TCP (Transmission Control Protocol) using an AC/DC Control Box. The control boxes of the two robotic arms are connected to a router via Ethernet cables, and the router is then connected to the host computer. Additionally, the two robot hands are directly connected to the same computer using RS-485 serial communication.

**Object Tracking.** Real-time object tracking is performed with an Intel RealSense D435 stereo camera. Since the object has a high color contrast from its background, we first use a simple color detector on the RGB image to find the pixel location of the object. The color range for detecting a blue object is constrained between $[80, 200, 0]$ and $[120, 255, 0]$ in HSV color space. Next, the 3D position of that pixel is obtained from querying the corresponding depth value on the depth image, where post-processing filters including disparity, spatial and temporal, are applied to reduce depth noise. Finally, we get the 3D object position in robot frame from image frame with calibrated camera extrinsics parameters.

**Domain Randomization** Isaac Gym offers several domain randomization functions for reinforcement learning training. We apply randomization to the task, as indicated in Table. 3 for each environment. We generate new randomizations every 1000 simulation steps.

| Parameter | Type | Distribution | Initial Range |
|---|---|---|---|
| **Robot** | | | |
| Mass | Scaling | uniform | [0.5, 1.5] |
| Friction | Scaling | uniform | [0.7, 1.3] |
| Joint Lower Limit | Scaling | loguniform | [0.0, 0.01] |
| Joint Upper Limit | Scaling | loguniform | [0.0, 0.01] |
| Joint Stiffness | Scaling | loguniform | [0.0, 0.01] |
| Joint Damping | Scaling | loguniform | [0.0, 0.01] |
| **Object** | | | |
| Mass | Scaling | uniform | [0.5, 1.5] |
| Friction | Scaling | uniform | [0.5, 1.5] |
| Scale | Scaling | uniform | [0.95, 1.05] |
| **Observation** | | | |
| Obs Correlated. Noise | Additive | gaussian | [0.0, 0.001] |
| Obs Uncorrelated. Noise | Additive | gaussian | [0.0, 0.002] |
| **Action** | | | |
| Action Correlated Noise | Additive | gaussian | [0.0, 0.015] |
| Action Uncorrelated Noise | Additive | gaussian | [0.0, 0.05] |
| **Environment** | | | |
| Gravity | Additive | normal | [0, 0.4] |

Table 3: Domain randomization parameters.

# B  Sim2Real Transfer

**System Identification.** To achieve a successful sim-to-real transfer, we utilize system identification techniques to align the behavior of the PD (Proportional-Derivative) controller of the arm and hand in simulation with that in the real world. This involves tuning the PD coefficients of the controllers to ensure that their responses to impulse and sinusoidal inputs are aligned. This step is crucial in ensuring that the control actions generated in simulation can be effectively applied to the real robot setup, enabling a reliable sim-to-real transfer of our system.

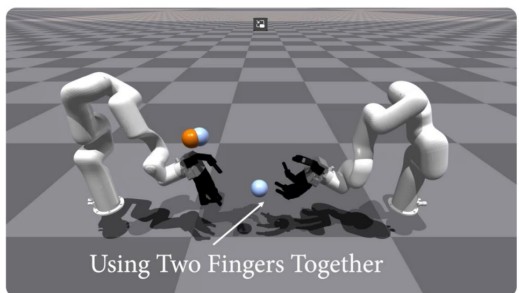
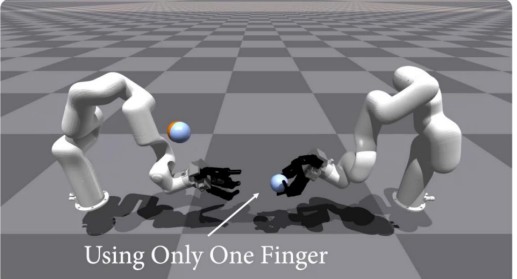

Using Two Fingers Together | Using Only One Finger

Figure 6: **Ablation study Left:** policy with rod in Training. **Right:** policy without Using rod in Training.

## C  Why is There a rod Used in Training Objects?

We find this leads to more robust policy in the training. Specifically, we observed that the simultaneous movement of all fingers when throwing a ball led to more robust results, whereas when using balls/square objects, policy often used only a few fingers. Therefore, we designed the shape of the rod in training. If the policy wants to throw the rod stably, it must learn to use all of its fingers in throwing, so it can make our policy more robust. We conducted new ablation experiment, the results of which are presented in Table.4. In terms of novel objects' success rate, we observed that the policy's success rate without using a rod is lower compared to when the rod is used. We have included videos on our website that showcases the hand's motion when training with and without the rod, as shown in Figure. 6. The video demonstrates that when the rod is excluded from training, the hand fails to effectively use all of its fingers during throwing, leading to a reduction in generalization ability.

| Ablation | Known obj | Novel obj |
|----------|-----------|-----------|
| Ours (w/o rod) | 0.94±0.04 | 0.26±0.06 |
| Ours | **0.95±0.03** | **0.37±0.04** |

Table 4: Ablation study in training w/w.o. rod

## D  How Pre-throw Conditions Impact Throwing Performance?

In this experiment, we examine the repeatability of the thrower. Our intuition is that a thrower policy capable of generating consistent object trajectories often results in a higher success rate for the catcher. We find that the robots' and objects' initial positions have a significant impact on the stability of the throwing motion. To address this, we investigate three different initial conditions (depicted in Figure 7) in real-world. For each initial position, we train a MARL policy and conduct 10 trials with the thrower robot. We compute the variance of the landing point on the table to evaluate the repeatability under different initial conditions. A smaller variance indicates better repeatability and stability. The results are summarized in Table 5. We observe that condition (c) demonstrates the smallest variance and enables more stable throws towards the target compared to conditions (a) and (b). This suggests that an initial firm grasp is advantageous for subsequent throwing behavior.

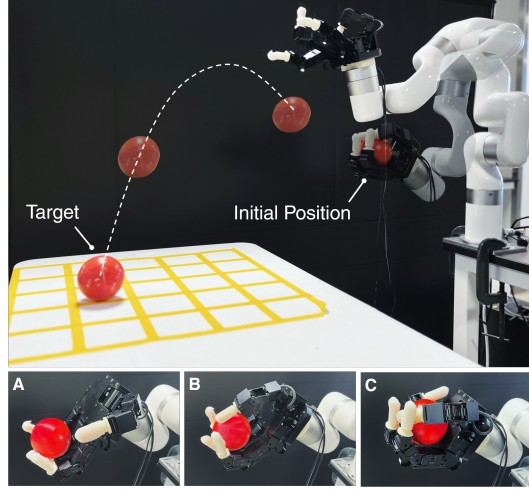

Figure 7: **Throwing Stability Test** of different initial settings: (a) simply placing the object on an open robot hand, (b) gripping the object with the robot hand, resembling a parallel gripper, and (c) firmly grasping the object with the robot hand.

| Settings | Pose A | Pose B | Pose C |
|----------|--------|--------|--------|
| Std(x) | 0.051 | 0.072 | **0.024** |
| Std(y) | 0.087 | 0.048 | **0.043** |

Table 5: **Comparison for Pre-throw Conditions:** We calculate the standard deviation of landing points on the table in the x and y directions, based on 10 runs for each pose. The units are in meters.

# E   Hyperparameters of the RL algorithms

Table. 6 and Table.7 are the hyperparameters of the RL algorithms.

| Hyperparameters | Throw and Catch |
|-----------------|-----------------|
| Num mini-batches | 1 |
| Num opt-epochs | 5 |
| Num episode-length | 8 |
| Hidden size | [1024, 1024, 512] |
| Use popart | True |
| Use value norm | True |
| Use proper time limits | False |
| Use huber loss | True |
| Huber delta | 10 |
| Clip range | 0.2 |
| Max grad norm | 10 |
| Learning rate | 5.e-4 |
| Opt-eps | 5.e-4 |
| Discount ($\gamma$) | 0.96 |
| GAE lambda ($\lambda$) | 0.95 |
| Std x coef | 1 |
| Std y coef | 0.5 |
| Ent-coef | 0 |

Table 6: Hyperparameters of MAPPO.

| Hyperparameters | Throw and Catch |
|-----------------|-----------------|
| Num mini-batches | 4 |
| Num opt-epochs | 5 |
| Num episode-length | 8 |
| Hidden size | [1024, 1024, 512] |
| Clip range | 0.2 |
| Max grad norm | 1 |
| Learning rate | 3.e-4 |
| Discount ($\gamma$) | 0.96 |
| GAE lambda ($\lambda$) | 0.95 |
| Init noise std | 0.8 |
| Desired kl | 0.016 |
| Ent-coef | 0 |

Table 7: Hyperparameters of PPO.

# F   Reward design

The reward of our system $r$ can be computed as $r = r_{dis} + r_{linvel} + r_{torque}$. In the design of our reward, $r_{dis}$ is the reward that mainly responds to throwing objects to the target position. $r_{linvel}$ is a reward that encourages throwers to release the ball from hand. $r_{torque}$ is a penalty item for robots that torque is too big. In our reward function, if $r_{dis}$ is missing, the object will not be thrown to the exact position, but will only be thrown forward vigorously. Without $r_{linvel}$, it would often fall into a sub-optimal where the thrower holds the ball in its hand and doesn't release. $r_{torque}$ is a common reward term that allows robots to avoid jitter and large dangerous movements.

# G   Perturbation Test in Simulation

We have conducted experiments such as perturbation for the test set and show the video results in simulation (see our project website, under the Perturbation Test section). We add three environmental conditions, e.g., wind flow opposing the thrower, along the thrower, orthogonal to the throw, as we show in our website, as shown in Figure. 8. We show the strength of the wind in our videos (1st column under Perturbation Test. Our policy can still perform well in these scenarios, indicating that we have learned a robust policy.

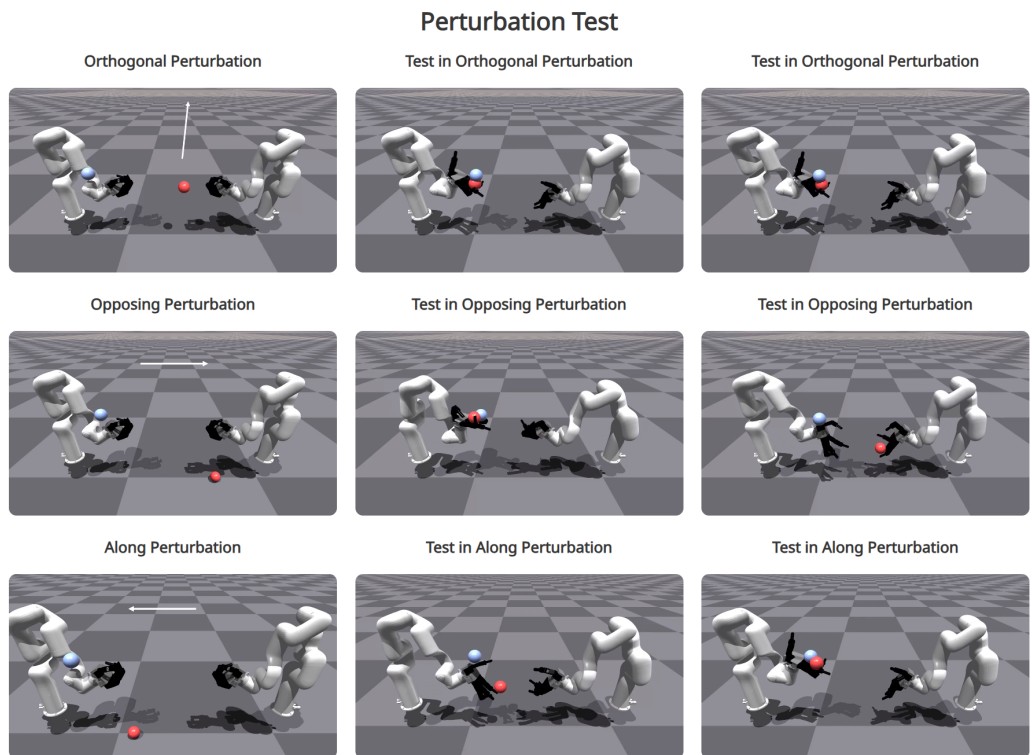

Figure 8: Perturbation test in simulation

