# OpenReview forum: "Dynamic Handover: Throw and Catch with Bimanual Hands"
_robot-learning.org/CoRL/2023/Conference — CoRL 2023 Poster_

### Official Review · Reviewer_GCGh · 2023-07-17

**Confidence:** 4
**Originality:** Good
**Technical Quality:** Very Good
**Clarity Of Presentation:** Very Good
**Impact:** 3

**Recommendation:**

Strong Accept: I recommend accepting the paper and will argue for my recommendation even if other reviewers hold a different opinion.

**Review:**

Overall comments: I think the authors have proposed an interesting problem in the bimanual manipulation and focus on the dynamic manipulation tasks that are less explored in prior works. The execution of the experiments (especially real robots) are well designed.


A high-level comment on the control and task design: Why is only delta positions enough? To my understanding, since only PD control is used here, the robot is only following delta positions. That being said, the control has nothing to do with the end effector velocities or acceleration, which are important to the dynamic manipulation authors focus on. While it might seem reasonable to do tasks like catch/throw in the paper, there are many dynamic manipulations that might be able to do with the current setup. I think this is a shortcoming in the proposed system which is supposed to solve a wide range of dynamic manipulation tasks. Following this point, there are two comments:
- Ln92: Please remove “Opening cap” example if this paper did not cover this task. It sounds a bit overclaimed.
- Ln 93: I think doing this requires “contact modes” that need to be achieved by more than one arm. Just having two arms do not constitute as necessary component for human-level skills. I think this claim should be rewritten in a more rigorous way.

Another shortcoming in this paper is its mismatch in the reported numbers between simulation and real-robot experiments --- Why does the simulation only have 28% success rate while all SRs in real-robot experiments are higher than 28%? I read that the simulation trail is evaluated over 100 while the real robots are evaluated over 5 trials. I think the statistical evaluations need to be improved.

**Quality Of The Limitations Section:**

Limitations are addressed clearly

**Questions For Rebuttal:**

- About Experiments: Why is there a rod (capsule object) used while none of the testing objects are similar to this object in geometry?
- Which point in the object is considered as its tracking point to specify its positions?
- My understanding is that the authors did not leverage the geometry information of objects. In this case, to what extent (in terms of size or shape variance) do we expect a learned policy to generalize?
- How do failure modes look like? Please include the videos.
- Why are the numbers in real robot experiments on unseen object generalization significantly higher than in simulation?

**Robotics Focus:**

Sufficient demonstration on hardware

**Summary Of Paper:**

This paper presents a method to learn a bimanual policy for a dynamic manipulation task, namely catch-and-throw. This paper formulates the task as a multi-agent problem and uses multi-agent RL to learn policies. The authors also designed a sim-to-real pipeline so that the RL policies can be cheaply learned in simulators and the policies can be deployed on the physical hardware. Both qualitative and quantitative results in simulation and on real robots are provided, and the learned policies are demonstrated to generalize on different objects that are never seen during policy training.

**Summary Of Recommendation:**

I think the paper is well-written, and the experiments are reasonably done. However, several questions need to be addressed (which I believe can be done during the rebuttal phase). I'll adjust the score based on the quality of the authors' replies (They don't need to be lengthy. I want to see the replies that are straight to the point.)

---

### Official Review · Reviewer_wGRc · 2023-07-18

**Confidence:** 4
**Originality:** Excellent
**Technical Quality:** Very Good
**Clarity Of Presentation:** Excellent
**Impact:** 4

**Recommendation:**

Strong Accept: I recommend accepting the paper and will argue for my recommendation even if other reviewers hold a different opinion.

**Review:**

### Strengths
- The result video is quite impressive and visually appealing. Setting up a complex real system like this one and getting sim-to-real to work as demonstrated should be a major achievement and not something commonly seen.
- The method is technically sound and well presented. The use of the goal position predictor seems sensible. For a dynamic and real-time task like this one, the model requires some kind of predictive capability for fast reaction.
- Overall, the paper is well written and a pleasant read. Graphics and illustrations are well used and helpful for understanding the content.

### Weaknesses
- It is a bit counter-intuitive how the proposed three-stage training pipeline can be expected to bridge the dynamics gap (L143-151). The goal estimator is still somewhat overfitting to the trajectories from the simulator, so how is that expected to improve the transfer?
- Following the last point, you fine-tune all the components with end-to-end training in stage 3. Will this further break the decoupling you deliberately established earlier between policy and goal estimation? Are there still some constraints for the goal estimator in stage 3?
- While the sim-to-real transfer looks impressive from the video, a thorough sim-to-real analysis is lacking. For example:
  - RL training has high variance, and you do multiple runs for each training. For each baseline, which run do you pick for real world evaluation?
  - Does the one performing well in sim (e.g., in terms of return) also perform well in real?
  - Can two models with similar training performance in sim have different qualitative performances in real and different sim-to-real transfer outcome?
  - For each baseline, do you test multiple models (e.g., trained from different runs) in real?
  - What are other sources of sim-to-real gap besides dynamics, e.g., perception? And how can you combat them?

### Questions and Suggestions
- [L23] "bring multiple challenges" -> "brings multiple challenges"
- [L103] "There are two robot agents in the this task: This task involves two robot agents:" -> Duplicated sentences.
- [Fig. 2] Why is object position 3 dimensional? Is it just (x, y, z) coordinates of the center?
- [Fig. 2] Since the thrower's policy also takes in target position, I suppose you are also randomizing the target goal position (otherwise you don't need that input)? However, I don't see this randomization mentioned anywhere. The throw goal seems to be pre-defined in stage 1 (L145: " the catcher observing the pre-defined throwing goal"). Then is it being randomized in stage 3?
- [L127] "For the XArm-6, we employ delta joint positions as the control target, while for the Allegro hand, we utilize absolute joint positions as the control target. This design choice is made to avoid jerky motion of robot arm for safety reasons, ..." -> Any reference or empirical evidence for avoiding jerky motion with this design?
- [L146] "(ii) Next, we freeze the base policy and train a goal estimator through supervised learning, using the rollout trajectory of the base policy as training data." -> Isn't the base thrower policy from (i) always consistently throwing to a pre-defined goal (see L136)? If so, how do you generate diverse trajectories to train the goal estimator?
- [L146] Following the last point, is the dynamics gap between sim and real coming from hand-object interaction, or also from the flying phase where the object is traversing though air passively subjecting to gravity force?
- [L148] "(iii) Finally, we replace the pre-defined throwing goal in the catcher’s observation with the estimated goal and unfreeze the policy for fine-tuning both the base policy and the goal estimator in an end-to-end fashion." -> Is there still a loss for goal prediction in stage 3? If no, will the goal estimator diverge from goal prediction with fine-tuning, i.e., will it learn to predict some arbitrary output as long as the policy can accomplish the final catching task?
- [L160] "..., the catcher's policy, ..., takes as input its proprioception, the pre-defined goal position ..." -> Should it be the "predicted goal position" instead of "pre-defined goal position"?
- [L165] "... we design the reward function using three components: (i) distance between object and throwing goal; (ii) object velocity projected in the direction from thrower to catcher; (iii) robot joint torque." -> Why is there no reward for catching?
- [Eq. 1] While easy to infer, please still explicitly explain what $\omega$ is.
- [Eq. 1] What is the meaning of superscript $1$ and $0$ here?
- [Tab. 2] "We calculate the standard deviation of ... We calculate the standard deviation in ..." -> Can be trimmed.
- [Fig. 5] For "Ours", is the training curve from stage 3?
- [L234] "... as we validate in Section 5.1" -> 5.2?
- [L236] "The utilization of multi-agent reinforcement learning and goal estimation proves beneficial for accomplishing the throwing and catching task." -> Why should "w/ MAPPO" outperform "w/o MAPPO"? The single agent setting should be strictly easier than the multi-agent setting, since both thrower and catcher have full access to the states of both sides. Therefore, the single agent setting should perform strictly better than the multi-agent setting? However, this result suggests otherwise which seems rather counter-intuitive.
- [L275] "table  2" -> "Table 2"
- [Supp L13] Object tracking is the real world may be noisy, while it is always noise-less in sim. Is the sim-to-real gap on that front taken into account? Do you need to add noise to object position in sim? Does this gap affect the goal estimator's performance in the real world since it takes the tracking output?
- [Supp L13] Following the last point, is there any bottleneck (i.e., frame rate) from the vision sensor? And how fast is the object tracking running? Is it faster than the control rate of the policy (i.e., in training)?

**Quality Of The Limitations Section:**

Limitations are addressed clearly

**Questions For Rebuttal:**

See the review above.

**Robotics Focus:**

Sufficient demonstration on hardware

**Summary Of Paper:**

- The paper aims to solve a dynamic and dexterous bi-manual handover problem, where one thrower robot is tasked to throw an object to a catcher robot, and the catcher should catch the object.
- Both robots have a multi-fingered hand mounted on a fixed base arm.
- A sim-to-real approach is adopted, where a policy is first trained in a simulated environment and then transferred to the real world.
- The task is treated as a two agent problem and the policy is trained with a multi-agent RL algorithm.
- The key idea is to use a goal estimator aside from the policy network to estimate the goal position of the thrown object for the catcher.
- To tackle the sim-to-real challenge, a three stage training pipeline is proposed, which trains the policy networks and goal estimator to be robust to variations in the target throwing position.
- The proposed method is compared against baselines in both sim and real, on 11 shape varying objects (in sim) and 3 sandbag objects (in real), respectively.

**Summary Of Recommendation:**

Overall, I think this is a solid work with good execution and impressive results. Some improvements can be made on the presentation of three-stage training and on the sim-to-real analysis. But this is already a strong submission regardless.

### Post Rebuttal
My initial review has been positive and the rebuttal does not change that view. I'm happy to keep my position and recommend a strong accept.

---

### Official Review · Reviewer_PiJr · 2023-07-19

**Confidence:** 4
**Originality:** Good
**Technical Quality:** Good
**Clarity Of Presentation:** Good
**Impact:** 3

**Recommendation:**

Weak Accept: I recommend accepting the paper, but will not argue for my recommendation if the majority of other reviewers have a different opinion.

**Review:**

Strengths:
- The problem is interesting, potentially useful (extending workspace), and difficult (two arms and hands, large action space, dynamic, sim2real)
- The experimental setup does not require a mocap system -- it uses a RealSense
- Training code is provided (although I did not run it)

Major weaknesses:
- I think multi agent RL for dexterous manipulation should not be listed as a contribution because of references [16] and [17]
- Dynamics randomization is not a contribution, since it's been a standard approach since about 2018 -- would recommend removing this from the contributions list
- The authors may have missed an important reference -- DexPBT by Petrenko et al. The work is simulation only but shows very impressive dynamic manipulation including between two hand + arm systems (see the videos on their project website)
- The real world objects are sandbags -- this does make them much easier to catch. This is acknowledged in the limitations section (low restitution).
- The simulation results are not particularly strong. There is a confusing result that the agent w/o MAPPO and goal estimation does better on known objects. (The training curves look noisy so I think it's worth trying this over more seeds and also training for more steps). Generalization to novel objects also gets a max success rate of 28%
- "this feature helps mitigate the impact of unpredictable parameters" -- does not seem to be a well supported statement. Why not vary these parameters directly in simulation to see if this is true?
- Success rates in the real world are quite low -- between 33-60%. I think this should be mentioned as early as possible in the paper (abstract and/or intro) so that readers are aware that this work is just a first step and doesn't reliably solve the problem
- No summary video of the paper is provided

Minor weaknesses:
- "Bimanual hands" is redundant -- bimanual means using both hands
- There are non standard phrases used and grammar errors throughout the paper -- would recommend closely revising the writing
- The thrower agent is fixed, the catcher agent is fixed and a throw + catch is executed once (not back and forth) -- only a minor weakness because just a single throw + catch is a good contribution
- Should mention that you use k prior states in figures 2 and 3
- Reward shaping is often a critical process -- would be good to get more insight on how you constructed the reward, what other rewards you tried, how sensitive the reward was to coefficients like 0.003 etc
- There is not much justification provided for the expanded set of simulated objects in figure 4b. Why these objects?
- MAPPO is critical to the paper and ablations but not much description of it (and differences with standard PPO) is provided
- Code uses outdated version of Isaac Gym
- Code is not documented properly (missing docs)

**Quality Of The Limitations Section:**

Additional details required

**Questions For Rebuttal:**

- What is meant by dynamic extrinsic dexterity?
- Can you please provide a comparison of methods and results between your work and DexPBT? DexPBT is simulation only but it would still be very useful to see a description of the pros/cons of your two very different methodologies as well as your simulation based results.
- How is the object tracked?
- The action space is said to be 22 dim per arm + hand but then it's said on line 131 that the system performs better when the robot joints are fixed -- please clarify what is the actual size of the action space then?
- How was the reward constructed and what did/didn't work well (see previous comment on reward)? I would also like to see an evaluation of the performance of the different reward formulations that were considered
- Please define the notation in equation 1. Related questions -- how is the predicted goal defined? Is it position after fixed number of timesteps? And is a neural net actually necessary to predict something as simple as goal location after a toss?
- It's not too clear to me why in phase 3 you simultaneously fine tuned both the estimator and the policy. I would like to request an ablation study for phase 3 -- simultaneous fine tuning of both the estimator and the policy versus just fine tuning the policy
- Can you train your simulated policies over more seeds and also for more timesteps to see if the trends from Table 1 (especially the surprising results for known objects) are still true?
- "after performing system identification" -- this seems to be a critical step. Can you describe your procedure and provide a comparison of performance with and without this procedure?
- It would be nice if there were 4 categories in Table 3 (the same ones as in Table 1) so that the individual effects of MAPPO and goal estimation can be seen
- Please provide videos of failure cases
- Is the pre throw configuration manually specified during training in simulation and testing on the real robot?
- What code do you plan to publicly release? The code to train the policies in simulation? The code to execute the policies on the hardware? Both?
- The policies appear to be all underhand throws and catches -- could you comment on this?

**Robotics Focus:**

Sufficient demonstration on hardware

**Summary Of Paper:**

The paper addresses the problem of throwing and catching objects from one hand + arm system to another. The robot behaviors are learned via multi agent RL in simulation and transferred to the real robot. Key contributions are the system itself (perhaps the first demo of sim2real for bimanual throwing/catching) and a potentially useful learning pipeline (training policies with a fixed final object state, training an estimator of the actual final object state, fine tuning the estimator and the policies).

From the methods side the use of multi agent RL is not novel (the key algorithm is MAPPO, which has been described before) and the use of multi agent RL for bimanual dexterous manipulation is also not novel (references [16] and [17] in the paper). However it seems that multi agent RL for dexterous manipulation + sim2real has not been done before.

Success rates are about 90% in simulation for known objects and 25-30% for unknown objects, and 33-60% in real for unknown objects.

**Summary Of Recommendation:**

The paper's core value is in demonstrating a system (maybe the first one) that uses sim2real for bimanual throwing and catching. In that sense the paper should be evaluated as a systems paper. The weaknesses on the systems side are the throwing/catching only happens once, there are only 3 real world objects, the success rates are low (33-60%), and although the real objects are technically unseen they are a bit easier to catch because they are bean bags. The video clips are nice to see since they demonstrate simultaneous coordination of many joints and some anticipatory and reactive behavior of the catching robot.

The paper's methodology is based on MAPPO and training an estimator of the final state of the thrown object. MAPPO is not new but the application to this context may be new. The estimator could potentially be claimed as contribution but I believe that the evaluations of its usefulness are weak in both simulation (unclear result for known objects) and real (limited ablation).

---

### Official Review · Reviewer_VRgh · 2023-07-21

**Confidence:** 4
**Originality:** Very Good
**Technical Quality:** Very Good
**Clarity Of Presentation:** Very Good
**Impact:** 4

**Recommendation:**

Strong Accept: I recommend accepting the paper and will argue for my recommendation even if other reviewers hold a different opinion.

**Review:**

The paper is well-written and the core ideas are presented comprehensively. Experimental results are impressive and I appreciate the detailed discussions on them. However, I believe that the manuscript could be improved by including/expanding on the following points:
- In Figure 1, left column, please explain why there are two blue balls in the scene.
- In Figure 3, please explain what is the violet ball in the first subfigure in the top row, what are the ball trajectory start and end points in the first figure in the bottom row, and what the red and blue balls indicate in the remaining figures. Please add explanations for these in the caption as well.
- Add details for the policy architecture used in the main text as well.
- Table 2, please specify the units used. Also, briefly describe the main take-aways.
- Table 3, in the caption, please specify again what HR and SR stand for for better readability. Also, briefly describe the main take-aways.
- Please explain how the ball velocity is estimated using the current setup.
- Please explain how the hyperparameters for the reward function were chosen.
- A comparison to the policies learned using kinesthetic teaching and motion primitives (DMPs/ProMPs) with goal estimation will help establish the proposed method strongly.
- It would be interesting to see how robust the learned policies are to environmental conditions, e.g., wind flow opposing the thrower, along the thrower, orthogonal to the throw, etc.

**Quality Of The Limitations Section:**

Limitations are addressed clearly

**Questions For Rebuttal:**

- It is stated in L162 that "..include observation from past k frames as input for both policies to provide temporal information..". While it is evident that adding last k frames should help the catcher to predict its goal position, it is unclear what benefits does it provide to the thrower policy. Please explain.
- I would like to see the generalization capabilities of the trained policies. I would be curious to know how the learned policies will perform if we swap the thrower and catcher robots.

**Robotics Focus:**

Sufficient demonstration on hardware

**Summary Of Paper:**

The paper presents a novel approach for throwing and catching objects with multi-fingered hands attached to robot arms using multi-agent reinforcement learning (MARL). In the proposed approach, thrower and catcher policies are trained using MARL in simulation and then deployed on the robot using Sim2Real transfer. A trajectory prediction model is also proposed to reduce the sim2Real gap and increase the chances of catching the object successfully. Experiments with multiple objects in a real-world setting demonstrate the superior performance of the proposed approach in comparison to baseline methods.

**Summary Of Recommendation:**

The paper is well-written and the core ideas are presented comprehensively. I enjoyed reading the detailed discussions on the experimental results. However, I am a bit concerned about the generalizability of the approach beyond the particular setup. It is unclear how well the learned policies will generalize to different settings, such as different embodiment or environmental conditions. If the authors can establish the generalization capabilities of the approach during the rebuttal period, I would feel more confident in recommending the paper for acceptance.


**Post-rebuttal update:**
The authors have addressed my concerns satisfactorily. I appreciate the additional experiments conducted by the authors to demonstrate the effectiveness of the proposed method. I have updated my recommendation accordingly.

---

> ### Author Response · Authors · 2023-08-15
> **Thank you for your comments**
>
> Dear reviewer,
>
> Thank you for your feedback and your comments again!
>
> Since tomorrow is the deadline for the rebuttal period, we wonder if you have feedback on our rebuttal response? Please let us know if you have any more questions.
>
> Best,
> Authors of Paper9

---

### Author Response · Authors · 2023-08-12
**General Comment by Paper9 Authors**

We thank all reviewers for their thoughtful comments. We have revised our manuscript based on your feedback (list of changes at the bottom of this comment), and responded to each of your individual comments.

We have updated our website to include both the summary video, new experiments results with new objects, and also failure cases. Please check [here](https://bimanualhand.github.io/).

In particular, we have also conducted new experiments in simulation where we include a wider range of objects with 11 training objects and 14 testing objects. The results are shown in the table below. We observe an improvement of generalization to novel objects (37%) as compared to 28% in Table 1 in the original submission. This suggests a large potential on the generalization ability of our approach when scaling up the training data.

|  New Simulation Ablation |Known obj  | Novel obj     |
| :---        |    :----:   |          ---: |
| w/o MA       | 0.89±0.07      | 0.24±0.05   |
| w/o GE | 0.88±0.04        | 0.22±0.04  |
| w/o Both        | 0.93±0.03      | 0.12±0.06   |
| Ours | **0.95±0.03**    | **0.37±0.04**  |

---

### Author Response · Authors · 2023-08-16
**[Confidential] About comparing to a RSS'23 paper**

Dear Area Chairs,

Thank you so much for serving as an Area Chair for CoRL. We have an active discussion with Reviewer PiJr, and we hope we have addressed all the reviewer's concerns. We want to bring your attention to reviewer PiJr listed that we did not compare to DexPBT as a major weakness. This is a RSS'23 paper and it is arxived half a month before the CoRL deadline. We believe this comment is a bit unfair. We also explained in our rebuttal that DexPBT is focusing on only doing sim experiments, and we perform sim2real for real robot experiments. We hope making the real robot work is a valuable point for the CoRL community.

Best,

Authors of Paper 9

---

### Decision · Program_Chairs · 2023-08-30

**Decision:**

Accept (Poster)

**Comment:**

The paper presents a system for dynamic handover using multi-agent RL and sim2real transfer. The reviewers agree that the presented task is interesting, useful, and challenging. Therefore, the result achieved by the final system is impressive.  In the rebuttal, the author provides additional experiment results that the reviewer finds informative, and we encourage the author to consider including this information in the final version of the paper.